# The *Foraging* Gene, a New Environmental Adaptation Player Involved in Xenobiotic Detoxification

**DOI:** 10.3390/ijerph18147508

**Published:** 2021-07-14

**Authors:** Marcel Amichot, Sophie Tarès

**Affiliations:** INRAE, Université Côte d’Azur, CNRS, ISA, Sophia Antipolis, F-06903 Valbonne, France; sophie.tares@inrae.fr

**Keywords:** foraging, adaptation, cGMP-dependent protein kinase, *Drosophila melanogaster*, insecticide, xenobiotic

## Abstract

Foraging is vital for animals, especially for food. In *Drosophila melanogaster*, this behavior is controlled by the *foraging* gene (*for*) which encodes a cyclic guanosine monophosphate (cGMP)-dependent protein kinase (PKG). In wild populations of Drosophila, rover individuals that exhibit long foraging trails and sitter individuals that exhibit short ones coexist and are characterized by high and low levels of PKG activity, respectively. We, therefore, postulated that rover flies are more exposed to environmental stresses, including xenobiotics contamination, than sitter flies. We then tested whether these flies differed in their ability to cope with xenobiotics by exposing them to insecticides from different chemical families. We performed toxicological tests and measured the activity and expression levels of different classes of detoxification enzymes. We have shown that a link exists between the *for* gene and certain cytochrome P450-dependent activities and that the expression of the insecticide-metabolizing cytochrome P450 *Cyp6a2* is controlled by the *for* gene. An unsuspected regulatory pathway of P450s expression involving the *for* gene in Drosophila is revealed and we demonstrate its involvement in adaptation to chemicals in the environment. This work can serve as a basis for reconsidering adaptation to xenobiotics in light of the behavior of species, including humans.

## 1. Introduction

Foraging is a vital behavior for animals to ensure their survival. However, this behavior is the result of a trade-off between the benefits, expenses, and risks associated with foraging. This has been formalized in the Optimal Foraging Theory which is based on several founding papers [1,2,3,4,5]. The perception of predation risk associated with foraging behavior has been extensively studied [6,7] and is a major factor in modulating this behavior.

The search for genes that control foraging behavior has been successful in *Drosophila melanogaster*. One set of strains is notable because a single locus was shown to control foraging and define rover and sitter populations [8]. Rover larvae continue to explore the environment even if they have found food while sitter larvae stop exploring when they are on food [9]. A single gene, *for*, has been associated with this locus. It encodes a cyclic guanosine monophosphate (cGMP)-dependent protein kinase (PKG) and its overexpression has been shown to be responsible for the foraging behavior [10]. In wild populations, this behavior is naturally polymorphic with a fairly constant proportion of rover and sitter flies (70% rovers, 30% sitters) [11] although this may vary with population density [12] or the allele frequency [13]. In addition, the *for* locus also controls adult dispersal [14]. Widely distributed in eukaryotes, PKGs have been described to modulate several physiological features such as behavior, memory, circadian rhythmicity, brain electrical activity, response to nociception, or axon guidance [15,16,17,18]. Surprisingly, it appears that PKGs perform identical or very similar functions in mammals.

During foraging and feeding, animals have to deal with chemical stresses caused by xenobiotics (natural or synthetic compounds found in the environment). Throughout evolution, they have been able to cope mainly by involving enzymes specialized in the excretion or degradation of these compounds. To all these stresses of natural origin, man has added pesticides. Their recent use has highlighted the rapidity and efficiency of the adaptive capacity of target organisms, namely pests. Indeed, pests challenge our chemical arsenal through a number of molecular means including modification of pesticide targets, increased excretion, or enhanced degradation of pesticides, resulting in resistance. Insects probably best illustrate the ability to develop such resistances, especially considering the increased degradation of insecticides [19,20]. The latter can be mediated by enzymes belonging to three categories: phase one enzymes (cleavage or inactivation of the compound), phase two enzymes (conjugation of the phase one metabolite or original compound to a nucleophilic molecule), and phase three enzymes (expulsion from the cell or excretion of the metabolites or original compound). Most phase one enzymes are esterases or cytochrome P450s, phase two enzymes are transferases (glutathione or UDP-glucuronosyl transferases) and phase three enzymes are ABC transmembrane transporters [21]. As far as we know, each category is present in all insect species, so that an insect may have dozens of P450s, GSTs, and ESTs [22].

Recent results in honeybees have shown that when intoxicated with a pyrethroid, λ-cyhalothrin, their degradation rate is the fastest at midday when the insect’s foraging activity is the most intense and is associated with increased expression of the detoxifying-P450 genes *Cyp9Q1* and *Cyp9Q2* [23]. Otherwise, previous studies had established that a higher level of *for* gene expression and PKG activities are detected in honeybee foragers when they performed tasks outside of the hive between noon and dusk and are then potentially more exposed to pesticides [24,25,26,27]. These data thus suggest that in honeybee foragers, a link may exist between foraging behavior controlled by the *for* gene and the activities of some P450s.

Here we wanted to test more precisely whether a link could exist between the phenotypes associated with the *for* gene in *D. melanogaster* (rover or sitter) and their ability to degrade or metabolize different xenobiotics (insecticides in this study). We, therefore, hypothesized that rover individuals, which travel longer distances and are thus potentially more exposed to various chemicals, would be more resistant and have a greater ability to degrade or metabolize them. The availability of mutants and transgenic flies of the *for* gene makes a very appropriate model for this study. The toxicological and metabolic fate of the insecticides we used here is also well known in this insect. Our results demonstrate that rover individuals are genetically better equipped to cope with insecticides and that the *foraging* gene plays a role in controlling the activity/expression of some, but not all, xenobiotic-metabolizing enzymes such as P450s.

## 2. Materials and Methods

### 2.1. D. melanogaster Strains

Two rover strains (*for*^*R*^, *dg2*-cDNA) and two sitter strains (*for^S2^*, *w^1^;for^S^*) were provided by Prof. MB. Sokolowski (University of Toronto, Canada). The *for^R^* strain was selected from a wild population by sorting rover and sitter larvae based on their foraging behavior [8]. The rover larvae were retained to establish the *for^R^* strain. After applying a mutagenic protocol on *for^R^* flies, a selection for a sitter phenotype was performed in the progeny to establish the *for^S2^* strain. It was demonstrated that the *for^R^* and *for^S2^* strains differ only in the *for* gene [8,28] so that *for^S2^* is hypomorphic for the *for* gene. The rover *dg2*-cDNA strain was derived from the sitter strain *w^1^;for^S^* after transgenesis using a *for* cDNA (T2 transcript) that is under the control of a leaky heat-shock protein promoter. Thus, the *dg2*-cDNA strain is a gain-of-function strain compared to *w^1^;for^S^* [10]. Because of this leakage and the fact that temperature interferes with toxicological measurements [29], we chose to avoid heat shock treatment. Thus, we had at our disposal two pairs of genetically related strains, *for^R^* and *for^S2^* on the one hand and *dg2*-cDNA and *w^1^;for^S^* on the other hand, differing only in foraging behavior associated with PKG expression level.

All flies were reared on standard cornmeal medium at 20 °C, 70% humidity, 12/12 h photoperiod. For every experiment, the same number of males and females was used.

### 2.2. Toxicological Tests

To test the adaptive ability of *D. melanogaster* strains, we selected three insecticides: aldrin, deltamethrin, and diazinon. These insecticides belong to different chemical families and have various targets in the nervous system. Aldrin is an organochlorine insecticide active on the GABA receptor, deltamethrin is a pyrethroid active on the voltage-dependent sodium channel and diazinon is an organophosphorus compound active on the acetylcholinesterase enzyme [30]. In addition, the metabolic fates of these insecticides and their toxicological consequences are well known in insects [31,32,33,34]. Indeed, these insecticides can be metabolized by ESTs, P450s, and GSTs. ESTs can cut ester bonds. P450s add an oxygen atom to the substrate, which then undergoes self-rearrangement to give the final product(s) of the reaction. GSTs catalyze the nucleophilic conjugation of the glutathione with molecules to make them excretable. All of these reactions generally result in less toxic compounds except for aldrin because its metabolite (dieldrin) is more toxic than the starting molecule (Figure 1).

The tarsal contact technique was used to test the response of *D. melanogaster* strains to insecticides. Glass tubes (20 cm^2^) were coated with 50 µL of an acetonic solution of the tested insecticide at different concentrations and allowed to dry in a fume hood for 2 h. Control tubes were treated with acetone only. Ten flies, males and females, were placed in each tube. These tests were repeated three times using 90 to 140 flies for each insecticide concentration and strain. For each insecticide, five concentrations were tested. Fly mortality was recorded after a 4 h period. We used Priprobit software (Priprobit 1.63 © Masayuki Sakuma) to calculate and statistically compare LC50s (concentration that kills half of the population expressed in µM) for each strain. In the tests, the P450s inhibitor piperonyl butoxide (PBO) (1 mM final) [35] was incorporated into 1 mL of cornmeal medium placed in the glass tube already coated with the insecticide solution.

### 2.3. Enzymatic Activities Measurements

The activities of ESTs, GSTs, and P450s were tested with substrates that are not specific to a single enzyme within a family but show global variations in the activity of a family. All these protocols, adapted to work with a single fly or with a single abdomen in microplates wells, were based on our experience or previously published protocols [36,37,38,39].

For all the enzyme activity measurements, flies were first anesthetized with CO_2_, sorted by sex, and then decapitated to remove eye pigments. The same number of males and females was used to avoid bias due to sexual dimorphism. To determine baseline values for each enzyme activity measurement, ten flies were placed in the appropriate reaction buffer (see below) and incubated at 100 °C for 10 min. Then, these flies were treated as described below for each enzyme family. Baseline values are subtracted from the relevant measurements to obtain activity values. For all measurements, we used 96-well microplates with a single fly or abdomen per well and activities were expressed in arbitrary units.

To measure esterase activities, each fly was crushed in 180 µL of HEPES (50 mM, pH 7.0) supplemented with 1 µL of bovine serum albumin (1 mg/mL). Twenty µL of substrate solution were added (alpha- or beta-naphthylacetate 1 mM in water/ethanol (99/1, *vol.*/*vol.*)). Incubation took place at 30 °C for 30 min; 20 µL of a fast garnet/SDS solution (10 mM each) were added to stop the reaction. The optical density of the samples was read with a Spectramax plus384 plate reader (Molecular Devices) at 550 or 490 nm for alpha- or beta-naphthylacetate derivatives, respectively.

GSTs activities were measured as follows: each fly was crushed in 50 µL of HEPES buffer (50 mM, pH 7.0). One hundred and fifty µL of glutathione (4 mM in HEPES 50 mM, pH 7.0) and 2 µL of monochlorobimane (30 mM in DMSO) were added. Control incubations were made either with heated flies, without fly, or without glutathione. Fluorescence (excitation: 390 nm; emission: 465 nm) was measured every 5 min for 40 min with a Cary Eclipse fluorescence spectrophotometer (Varian). We used at least 40 flies per strain in three different tests.

P450s activities were measured using 7-Ethoxy-Coumarin-O-Deethylase (ECOD). Freshly cut abdomens were individually placed in a well containing 50 µL of phosphate buffer (50 mM, pH 8.0) and 1 µL of bovine serum albumin (1 mg/mL). The reaction was started by adding 50 µL of phosphate buffer and 1 µL of ethoxycoumarin (20 mM in methanol). Three types of controls were made: wells without abdomens, wells containing one abdomen supplemented with PBO (5 µL of a 100 mM solution), and wells containing one heated abdomen. Fluorescence was measured after incubation for 2 h at 30 °C and the addition of 100 µL of a *v*/*v* mixture of glycine (100 µM, pH 10.4) and ethanol (excitation: 390 nm, emission: 450 nm). The tests were repeated 3 times and measurements were performed using 24 flies per strain.

We compared the enzyme activity levels pairwise with the Wilcoxon test using the R software (www.R-project.org (accessed on 7 February 2019)).

### 2.4. Northern Blotting

Total RNAs were extracted using Trizol^®^ (Invitrogen) according to the manufacturer’s protocol. Electrophoretic separation, visualization, and RNA blotting were performed according to http://www.protocol-online.org/cgi-bin/prot/view_cache.cgi?ID=787 (accessed on 16 July 2007). Probes were labeled using the PCR DIG probe synthesis kit from Roche. The sequences of the oligonucleotides used to synthesize the probes are listed in Table 1. Primer sequences for the P450s *Cyp6a2* and *Cyp6g1* were designed using AmplifX 2.0.7 (Nicolas Jullien; Aix-Marseille Univ, CNRS, INP, Inst Neurophysiopathol, Marseille, France—https://inp.univ-amu.fr/en/amplifx-manage-test-and-design-your-primers-for-pcr (accessed on 6 January 2020). Hybridization and signal detection were performed under high stringency conditions according to Roche specifications.

### 2.5. Accord Insertion Detection

An *Accord* transposable element inserted at the 5′ end of *Cyp6g1* gene was previously identified to be linked to overexpression of this gene in several resistant fly strains [40]. PCRs used to detect the *Accord* insertion were then performed as described in [40]. Using the primers listed in Table 1. Genomic DNAs of the different fly strains were extracted according to the Vienna Drosophila Resource Center protocol available at (https://stockcenter.vdrc.at/images/downloads/GoodQualityGenomicDNA.pdf (accessed on 16 February 2018).

## 3. Results

### 3.1. Foraging and the Response to Chemical Stresses in D. melanogaster

To test whether the *for* gene could modify the response to chemical stresses, we calculated the LC50 of rover and sitter strains for three insecticides belonging to three different chemical classes and with different targets (Figure 2).

With the organochlorine aldrin, we observed that rover flies were more susceptible than sitter flies in both strain pairs. We found the qualitatively opposite result with the pyrethroid deltamethrin, the rover strains being more resistant than sitter strains. With the third insecticide, the organophosphorus diazinon, we were unable to differentiate between rover and sitter strains in any pair. Thus, the responses of strains to insecticides are related to their behavioral status.

### 3.2. Xenobiotics Degradative Enzymes

We wanted to test whether the toxicological differences observed between the strains could be related to a variation in one of the enzyme activities known to be responsible for insecticides metabolization [41]. Of the activities we tested, those of alpha- or beta-esterases and glutathione-S-transferases activities did not vary significantly between the rover and sitter flies within each strain pair (Figure 3).

In contrast, rover flies *for^R^* and *dg2*-cDNA exhibited significantly higher ECOD activity than sitter flies *for^S2^* and *w^1^;for^S^* (Figure 3). Thus, this P450s-related activity is a coupled feature of fly behavior.

### 3.3. Cytochrome P450s Activities, Foraging, and Toxicology

We performed toxicological tests in the presence of a P450s inhibitor, PBO, to verify that P450s are indeed involved in the toxicological differences we observed between rover and sitter flies. To do this, we fed PBO to *for^R^* and *for^S2^* flies, the most different rover and sitter flies in terms of toxicology and ECOD activity, and performed toxicological tests with aldrin, deltamethrin, and diazinon (Figure 4). This resulted in the disappearance of the toxicological differences that we had previously recorded because the LC50s obtained are now similar between *for^R^* and *for^S2^* in the presence of PBO regardless of the insecticide. We, therefore, demonstrate here that P450s are directly involved in the differences in resistance levels observed between these two strains. Thus, we establish a link between toxicological characteristics, P450 activity levels, and the foraging behavior in these *D. melanogaster* strains.

### 3.4. Cytochrome P450 Genes Expression

In *D. melanogaster*, several P450 genes have been linked to insecticide resistance, but we focused our work on two P450s that were found to be insecticide metabolizers: *Cyp6a2* and *Cyp6g1* [42,43]. We compared their expressions by Northern blot in rover and sitter strains.

Figure 5A shows that *Cyp6g1* expression is significant but similar in *for^R^* and *for^S2^*. We tested by PCR for the presence of an *Accord* element insertion in the *Cyp6g1* promoter sequence as it has been shown previously to be responsible for overexpression of this P450 [44]. Both *for^R^* and *for^S2^* strains are positive for this assay (Figure 5A). In contrast, there is neither detectable expression of *Cyp6g1* nor insertion of the *Accord* element in the *dg2*-cDNA and *w^1^;for^S^* strains.

The situation is different for *Cyp6a2* as it is clearly expressed at a higher level in both rover strains *for^R^* and *dg2*-cDNA than in sitter ones *for^S2^* and *w^1^;for^S^* (Figure 5B). Thus, the expression level of *Cyp6a2* is clearly related to the foraging behavior.

## 4. Discussion

Rover individuals, which travel long distances to feed, are potentially more exposed to various stressors or chemical compounds than sitter individuals. We, therefore, have hypothesized that a link could exist between the foraging behavior driving by the *for* gene and the ability of insects to degrade or metabolize different xenobiotics. The toxicological analysis of the genetically related strains, *for* and *dg2*-cDNA (rover) on the one hand and *for^S2^* and *w^1^;for^S^* (sitter) on the other was the first step to test our hypothesis. Contrasting results were obtained as rover flies were more susceptible to aldrin but more resistant to deltamethrin while there was no difference between strains for diazinon. The effect of the *for* gene, therefore, seems to be complex as it can be positive (resistance), negative (susceptibility), or neutral depending on the chemical structure of the stressor. To explain these results, we first examined the relationships between the modes of action of these insecticides and their molecular targets and PKG pathway. The molecular targets of aldrin, deltamethrin, and diazinon are the GABA-gated chloride channel (*Rdl*), the voltage-gated sodium channel, and the acetylcholinesterase, respectively [30]. Of these three proteins, only *Rdl* is known to interact with the PKG. The expression level of *Rdl* is under the control of the PKG [45] and *Rdl* can also be phosphorylated by the PKG [46]. We have no other evidence to link the toxicological characteristics of the rover and sitter strains to PKG and molecular targets of insecticides. Furthermore, the cellular location of PKG (intracellular) and acetylcholinesterase (extracellular in the synaptic cleft) does not favor interaction between these two proteins. Thus, an interaction between PKG and these insecticide targets is very unlikely and therefore cannot explain the modulation of fly adaptive ability to xenobiotics.

We then tested whether insecticide metabolization could explain the toxicological results. So, the activities of EST, GST, and P450 enzymes were measured. Because ESTs and GSTs activity levels were not able to clearly differentiate rover from sitter flies, we concluded that they were not significantly involved in the adaptive process we studied here. In contrast to these observations, the ECOD activities carried by P450s here significantly higher in rover strains (*for^R^* and *dg2*-cDNA) than in sitter ones (*for^S2^* and *w^1^;for^S^*). The known modes of action of P450s on the insecticides we used could explain the toxicological characteristics of rover and sitter strains. Indeed, deltamethrin is inactivated by P450s to 8-hydroxy-deltamethrin which is no longer toxic [34], making rover flies more resistant. Aldrin is oxidized by P450s to the more toxic dieldrin [33] so rover flies should be more susceptible. Finally, diazinon is metabolized by P450s to either diazoxon (highly toxic) or diethyl-phosphate plus 2-isopropoxy-4-methyl-6-pyrimidine (non-toxic) [32], which should result in balanced toxicological tests. Our results are fully consistent with this scheme. To confirm the role of P450s, we tested the effect of an inhibitor of these enzymes, PBO, which then abolished the toxicological differences we observed between the rover and sitter flies in the presence of aldrin or deltamethrin, confirming the role of P450s in the toxicological profiles of the rover and sitter flies. Therefore, the *for* gene is involved in the control of the adaptive ability of flies mainly through the control of P450s activity levels, and the impact of this regulation is positive or negative depending on the xenobiotic encountered.

Because of the involvement of the two P450s *Cyp6a2* and *Cyp6g1* in metabolic resistance to insecticides in *D. melanogaster*, we traced their expression levels in rover and sitter strains. The *Cyp6g1* gene was overexpressed in *for^R^* and *for^S2^* compared to *dg2*-cDNA and *w^1^;for^S^*, but this feature is not consistent with the toxicological profiles or enzymatic activities observed in these strains. We also showed that the *Accord* transposable element insertion was present in *for^R^* and *for^S2^* at the site previously shown to induce *Cyp6g1* overexpression [44]. From all of these data, we concluded that *Cyp6g1* expression levels were independent of PKG function and thus not involved in the variations in fitness that we observed between rover and sitter flies. In contrast, we found the *Cyp6a2* gene overexpressed in rover flies *for^R^* and *dg2*-cDNA. In addition, we knew that CYP6A2 was able to metabolize ethoxycoumarin as well as diazinon, aldrin, and deltamethrin ([45], Amichot et al., unpublished data). Although we did not demonstrate that *Cyp6a2* was an effector of adaptation to the *for*-controlled environment, we demonstrated here that *for* was able to modulate the expression level of this P450.

Links between PKG and the regulation of P450s expression have been documented in the literature. Previous works with mouse primary hepatocyte cultures [47,48] suggested that PKG was involved in phenobarbital-induced upregulation of CYP2b9/10 or CYP3A expression. A recent publication demonstrated that PKG can regulate the function of *Nrf2* [49], a gene known to control the expression of P450s [50] and more specifically *Cyp6a2* [51]. As *Cyp6a2* expression has previously been shown to be controlled by phenobarbital [52,53], all these published data are consistent with our current data highlighting a direct link between PKG and the expression of the P450 *Cyp6a2*.

Furthermore, interactions between environmental variations and PKG have already been documented in *D. melanogaster*. PKG plays a role in thermotolerance [54,55] and resistance to anoxia [56]. Interestingly, another study identified polymorphic regions of the *D. melanogaster* genome in relation to environmental adaptation [57] and *for* was one of the genes identified as polymorphic in relation to environmental selection pressure (temperate vs. subtropical climate). To all these studies we now add a link between the *for* gene and xenobiotic tolerance making it a major player in the adaptation of *D. melanogaster* to its environment.

Such a relationship can have important ecological consequences. Here we have shown that a single gene can have opposite effects on the adaptive ability of the flies depending on their foraging behavior and the compound they face. Since a natural population is composed of rover and sitter flies [12,13], we can assume that population survival should be possible regardless of the compound present in the environment.

## 5. Conclusions

We have highlighted here a novel relationship between foraging behavior and adaptation to xenobiotics. Specifically, this adaptability is related to P450s activities which are themselves modulated by foraging behavior. It was previously thought that only direct induction by xenobiotics could affect the overexpression of such P450s. The relationship that we have described here between P450s and foraging behavior is therefore original and deserves to be studied further to identify the genetic mechanisms involved. What might be the ecological consequences of such a relationship? Are only foraging individuals involved in the development of insecticide-resistant pest populations? Wouldn’t it be enough for individuals to be under chemical stress to become resistant, but should they also be genetically predisposed? Can we then predict if a population can develop resistance by identifying the alleles of the *for* gene it contains?

PKGs have many similar physiological functions between *D. melanogaster*, and other insects and mammals, including humans. However, to what extend? A recent study described a polymorphism in the human PRKG1 gene (ortholog of the *for* gene) responsible for foraging behavior similar to that observed in the adult sitter fly. Thus, carriers of the PRKG1 AA genotype adopted a known risk-minimizing foraging strategy of moving to the periphery of the test arena [58]. Did they have different or even reduced xenobiotic/drug-metabolizing abilities compared to noncarriers of this genotype? Is the link between the for gene and the ability to adapt to its environment the same in mammals?

## Figures and Tables

**Figure 1 ijerph-18-07508-f001:**
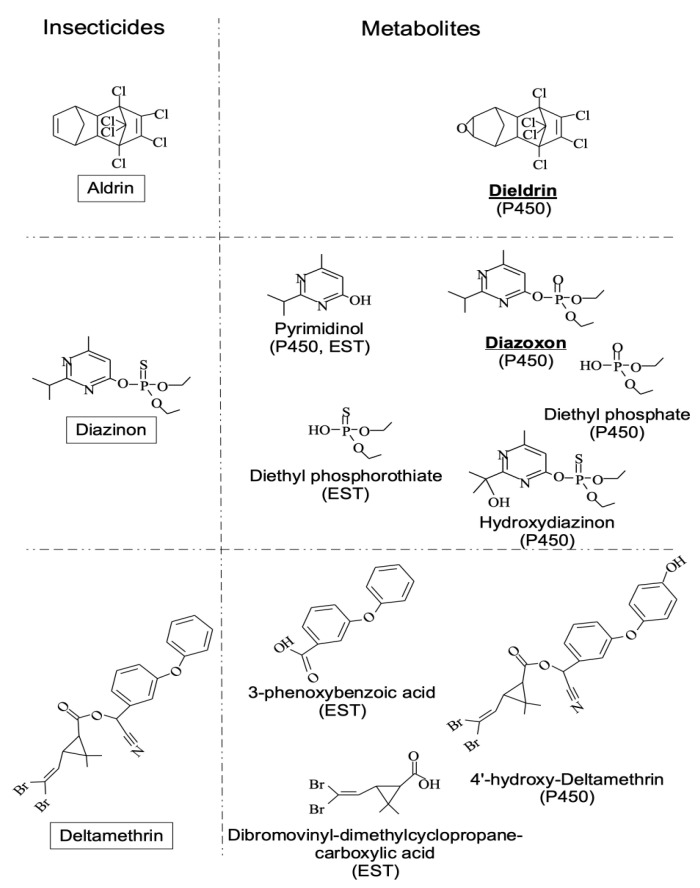
Main EST and P450 metabolites of aldrin, diazinon, and deltamethrin in insects.

**Figure 2 ijerph-18-07508-f002:**
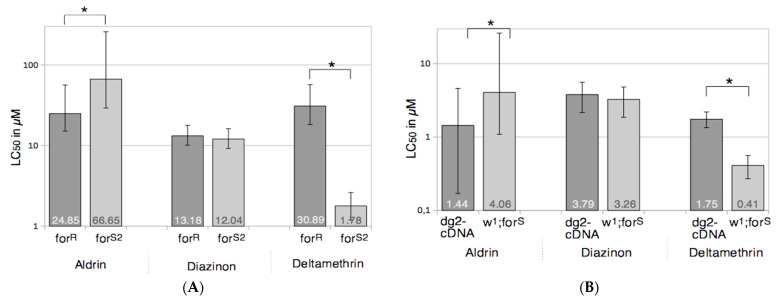
LC50 values of rover and sitter strains for three insecticides. Bar charts represent the LC50 values of the *for^R^* and *for^S2^* strains (panel **A**) and the *dg2*-cDNA and *w^1^;for^S^* strains (panel **B**). LC50 values are shown in the bars and are expressed in the same units as the insecticide concentrations of the acetonic solutions used to coat the tubes. Vertical bars represent 95% fiducial limits of the LC50 values. Dark grey boxes indicate rover strains and light grey boxes indicate the sitter strains. * indicate significantly different LC50 values.

**Figure 3 ijerph-18-07508-f003:**
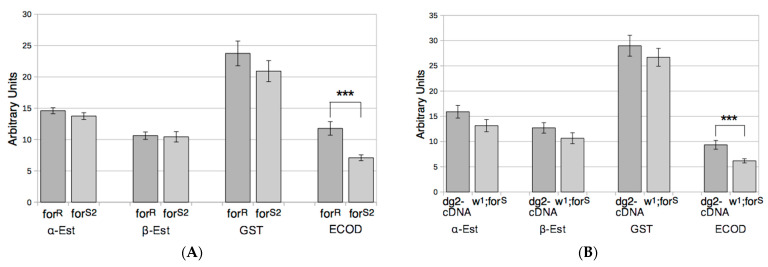
Activity levels of sterases, GSTs, and ECOD. Esterase activities are expressed in arbitrary units of O.D./fly/30 min, GSTs activity is expressed in arbitrary units of fluorescence/abdomen/40 min and ECOD activity is expressed in arbitrary units of fluorescence /abdomen/120 min. Results from genetically related strains are grouped in panel (**A**) (*for^R^* and *for^S2^*) and panel (**B**) (*dg2*-cDNA and *w^1^;for^S^*). The vertical bars represent the standard error of the measurements. *** indicate significantly different activity values at *p* < 0.005. Dark grey boxes indicate rover strains and light grey boxes indicate sitter strains.

**Figure 4 ijerph-18-07508-f004:**
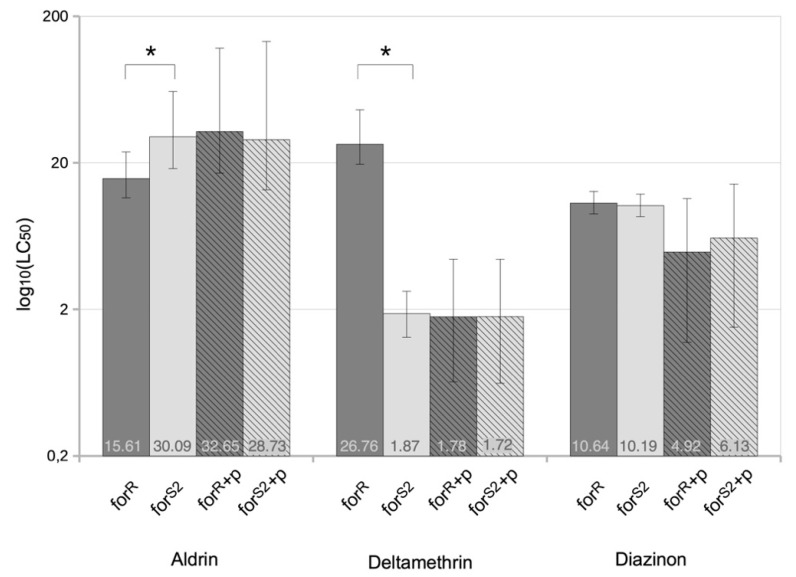
Inhibition of P450s and tolerance to insecticides. LC50 values are shown in the bars. Hatched boxes indicate that flies were treated with piperonyl butoxide. Vertical bars represent 95% fiducial limits of LC50 values. Dark grey boxes indicate rover strains and light grey boxes indicate sitter strains. * indicate significantly different LC50 values.

**Figure 5 ijerph-18-07508-f005:**
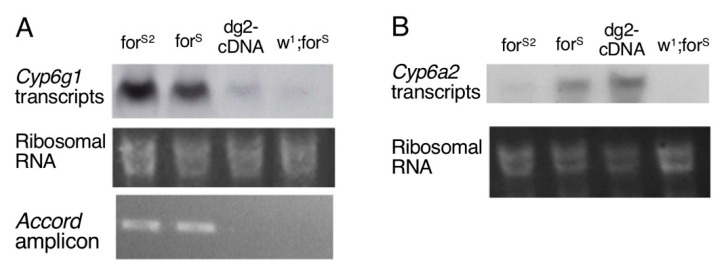
Expression profiles of *Cyp6g1* and *Cyp6a2.* (**A**) Northern blot detection of *Cyp6g1* transcripts and PCR detection of the *Accord* insertion in the *Cyp6g1* promoter. (**B**) Northern blot detection of the *Cyp6a2* transcripts. In both panels, the 18S RNAs used as controls show similar RNA loading in the lanes.

**Table 1 ijerph-18-07508-t001:** PCR primers sequences.

Gene	Forward	Reverse
*Cyp6a2*	CTGGTCAACGACACGATTGC	GTAGGTCATGGCCTTGATGG
*Cyp6g1*	CGATCATTGCAACACCAAGG	TCGCGTATTATCAAGCCGGG
*Accord*	GGGTGCAACAGAGTTTCAGGTA	CTTTTTGTGTGCTATGGTTTAGTTAG

## Data Availability

The data that support the findings of this study are available from the corresponding author upon reasonable request.

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
