# Peer review of "The *Foraging* Gene, a New Environmental Adaptation Player Involved in Xenobiotic Detoxification"

_ijerph, 2021, doi:10.3390/ijerph18147508_

Round 1
Reviewer 1 Report
The manuscript is a well written article and will provide good value to the audience. Please consider minor English formatting and thorough spell check before final submission for publication.
Author Response
Dear reviewer,
Thank you very much for your time reviewing our manuscript. In order to check the English language, we submitted our manuscript to a dedicated structure at our Institute. In addition, we modified our manuscript taking into account all the reviewers’ recommendations. We believe that these modifications have improved the quality of the manuscript.
Yours sincerely,
M Amichot and S. Tarès
Reviewer 2 Report
The manuscript written by Amichot and Tares entitled “The Foraging Gene, a New Environmental Adaptation Player 2 Involved in the Detoxification of Xenobiotics” reports some interesting results. The research article describes the link between for gene and P450 in Drosophilla melanogaster. However, the authors need to revise the manuscript carefully before considering further.
Suggestions/comments:
- cGMP please write the full name first time then use abbreviation. Check throughout the manuscript for others.
- Authors need to revise the confusing sentencesat line 45-49.
- The line no. 60-69 needs to be rephrased and rewritten.
- (forR, dg2-cDNA) at line. 70 please check it.
- Line no. 85 and 86 compiles together.
- melanogastershould be italics throughout the manuscript.
- Please do not use full name of Drosophila melanogaster in later pages and instead of it use melanogaster.
- Introduction is easy to read but needs a little completed.For example, “some of the enzymes active against xenobiotics are also able to metabolize allelochemicals and that was actually their primary function”. Please added more references to support the statement, and more details about the enzymes, how about they work? Please also add more information about the toxicity of used allelochemicals in introduction part which authors describing here in present study. This way the authors will demonstrate that they really have a good knowledge of the related literature. The following papers belong to this topic, I hope they are useful for you. doi: 1016/j.jhazmat.2020.125026; doi:10.1016/j.biortech.2020.123845; doi: 10.1016/j.chemosphere.2019.125507.
- Revise the line no. 122- 124.
- Merge the section 2.4 with section 2.3.
- Unnecessary spacing in Northern blotting section.
- It is not clear what is the “Accord insertion detection”???
- Figure legends are not with high resolution please revise the figures for all the mistakes
- The discussion section of the article was not written well. Please revise the discussion section with more recent articles and add more information with critically comparative analysis with your results.
- Revise the conclusion section as per the findings and how it can be useful for future research and improvement of life quality at larger scale.
Author Response
Dear Reviewer,
Thank you very much for your comments, here are our responses to each point :
1/ Corrected as recommended,
2/ These sentences have been rewritten,
3/ These lines have been rewritten. These changes were actually included in the changes we made in the Introduction section,
4/ The names of the fly strains have been verifed and are correct,
5/, 6/ and 7/Corrected according to recommendations,
8/ We have modified the Introduction section to make it clearer by removing all references to allelochemicals, as this was confusing. In fact, our manuscript focuses on xenobiotics, not allelochemicals. We also added the first suggested reference but not the other two because their subject (bioremediation) was too far removed from that of our manuscript,
9/, 10/ and 11/ Corrected according to recommendations,
12/ We have rewritten the " Accord insertion detection " part of our manuscript by clarifying what is “Accord insertion “ and by integrating appropriate reference,
13/ We hope to have improved the resolution of figures, slightly modified Figure 5 and corrected the legends,
14/ We have carefully rewritten the Discussion and introduced additional references,
15/ We have rewritten the Conclusion and added questions which can be seen as suggestions for future researches.
We also submitted our manuscript to a dedicated structure at our Institute to improve the English language.
Hoping to have met your expectations.
Yours sincerely,
M Amichot and S. Tarès
Reviewer 3 Report
Congratulations to the authors. I only think that should improve the introduction in terms of previous research or results similar to the author's research.
Author Response
Dear reviewer,
Thank you very much for reviewing our manuscript. Following your recommendations, we have modified the introduction by adding a some references and deleting some sentences and terms that we no longer found appropriate. We also checked the English language with the help of a dedicated structure at our Institute.
Yours sincerely,
M Amichot and S. Tarès
Round 2
Reviewer 2 Report
The authors have considered all comments raised by the reviewer and revised the manuscript accordingly based on these comments. The revision is fine and can be accepted for publication in current form.